# Placental, Foetal, and Maternal Serum Metabolomic Profiles in Pregnancy-Associated Cancer: Walker-256 Tumour Model in a Time-Course Analysis

**DOI:** 10.3390/ijms241713026

**Published:** 2023-08-22

**Authors:** Carla De Moraes Salgado, Laís Rosa Viana, Maria Cristina Cintra Gomes-Marcondes

**Affiliations:** Nutrition and Cancer Laboratory, Department of Structural and Functional Biology, Institute of Biology, University of Campinas, Sao Paulo 13083-862, Brazil; c104714@dac.unicamp.br

**Keywords:** cancer during pregnancy, metabolism, placenta, foetus

## Abstract

Cancer during pregnancy presents a delicate coexistence, imposing ethical and professional challenges on both the patient and medical team. In this study, we aimed to explore in a pre-clinical model the impact of tumour evolution in serum, placental and foetal metabolomics profiles during pregnancy in a time-course manner. Pregnant Wistar rats were distributed into two experimental groups: Control (C) and Walker-256 tumour-bearing (W). The rats were euthanised on three different gestational periods: at 12 days post-conception (dpc), at 16 dpc, and at 19 dpc. Serum, placenta and foetal metabolomic profiles were performed by ^1^H-NMR spectra following the analyses using Chenomx NMR Analysis Software V8.3. The tumour evolution was exponential, affecting the placental metabolomic profile during all the pregnancy stages. The placental tissue in tumour-bearing dams developed at a lower speed, decreasing the foetus’s weight. Associated with the serum metabolomic changes related to tumour growth, the placental metabolomic alterations impacted many metabolic pathways related to energy provision, protein synthesis and signalling, which directly harmed the foetus’s development. The development of the foetus is clearly affected by the damage induced by the tumour evolution, which alters the metabolic profile of both the serum and the placenta, impairing early embryonic development.

## 1. Introduction

The incidence of pregnancy during cancer evolution is relatively low, affecting about 1 in 1000 pregnancies [1]. This condition is characterised by the diagnosis of cancer during pregnancy, while breastfeeding, or within one year after delivery [2]. Although the incidence is relatively low, the coexistence between the patient and medical team requires delicate handling and poses ethical and professional challenges. The focus should be on ensuring the health of both the mother and the foetus [3]. It requires a multidisciplinary and individualized approach, considering the risks and benefits of treatments for both patient and baby [4,5]. The presence of a tumour during pregnancy can result in complications like intrauterine growth restriction (IUGR), premature birth, or perinatal death [5]. It is possible that impaired foetal development is linked to disruptions in placental function caused by tumour evolutions, which can alter the mother’s metabolic serum levels [6,7,8,9].

The placenta plays a crucial role in maintaining the health of pregnancy and foetal development, and its effectiveness is determined by its interaction with the mother’s responses and vice versa [10]. Overall, it is clear that the negative impact of cancer affects the metabolism and function of the placenta [11,12]. Despite the harmful effects of tumour growth in the context of pregnancy, there have been limited studies on the factors that cause these changes in both placenta and the foetus [11]. There is a significant public health concern regarding the link between cancer and pregnancy, which highlights the necessity for additional research in this area. In this context, it is important to study the interaction and variation of metabolites among the serum, placenta and foetus, using the “omics” technologies, which could provide a valuable understanding of the effects of cancer on pregnancy over the metabolic function [13].

Therefore, the purpose of this study was to explore how the progression of tumour evolution impacts the serum, placental and foetal metabolomic profiles during pregnancy in a time-course manner. As far as we know, this is the first study to investigate how changes in maternal serum and the placental metabolic profile, led by the harmful effects of tumour evolution, could reflect in impacted metabolic pathways of the developing foetus at three distinct stages of pregnancy.

## 2. Results

### 2.1. Morphometric and Metabolomic Profile during the Course of a Healthy Pregnancy

As expected, we noticed a significant increase in placental weight during the course of pregnancy, with the most significant increase occurring between 12 days post-conception (dpc) and 16 dpc (Figure 1a). During the healthy pregnancy (control group), the weight of the foetus steadily increased; with the most significant increase happening between 16 dpc and 19 dpc. (Figure 1b). The ratio of foetal and placental growth (F/P) increased during the three periods, with the largest increase observed between 16 dpc and 19 dpc (Figure 1c). The maternal morphometric data are described in Appendix A.

In this study, we also identified and quantified 51 metabolites in the serum, 59 in the placental tissue, and 61 in the foetal tissue in both pregnant groups (control and tumour-bearing groups). The detailed list of all the found metabolites and their differences are shown in Appendix A (Appendix A (serum results), Appendix A (placental results), and Appendix A (foetal data)).

During the healthy pregnancy, from the 51 serum metabolites found, 45 of them exhibited significant changes throughout the three different stages of pregnancy (Appendix A). These metabolite variations were related to amino acid metabolism (biosynthesis and mobilization) and the energy source (Appendix A).

The placental tissue analysis revealed that there were significant differences in 34 of the 59 metabolites found, throughout the three stages of a healthy pregnancy (C) (Appendix A). Among all the metabolites in the placenta of the control group, we observed that in a time-course analysis, almost all the placental metabolites were slightly reduced at both 16 dpc and 19 dpc compared to 12 dpc, except for some metabolites that had a linear content as the main energetic metabolites—lactate, nicotinurate and succinate (Appendix A). We identified the metabolic pathways affected during a healthy pregnancy by analysing the list of altered metabolites (Appendix A).

Out of the 61 foetal metabolites identified, 30 of them displayed significant variations across the three stages of pregnancy in the C group. (Appendix A). In a time-course analysis, we observed that 9 foetal metabolites were slightly reduced at 16 and 19 dpc vs. 12 dpc, and 8 foetal metabolites were slightly increased at 16 and 19 dpc vs. 12 dpc, 8 foetal metabolites slightly reduced at 19 dpc vs. 16 dpc and 10 foetal metabolites slightly increased at 19 dpc vs. 16 dpc (Appendix A). The serum, placental and foetal metabolic pathways throughout the healthy pregnancy are listed in Appendix A.

Appendix A display representative ^1^H NMR spectra of the serum, placental and foetal tissues, respectively, showing the respective metabolite identification.

### 2.2. Walker-256 Tumour Evolution Jeopardized Foetal Growth

The evolution of the Walker-256 tumour during pregnancy affected the placental and foetal weight compared to the control group (Figure 1). The development of placental tissue in tumour-bearing rats was slower than in the control ones, despite not being statistically different (Figure 1a). The foetal weights of the W group were lower from 16 dpc to 19 dpc. Additionally, the ratio of foetal weight to placental weight (F/P) was also significantly lower in the W group compared to the control group. (Figure 1b,c). Throughout the course of pregnancy, we observed that the tumour’s growth followed an exponential pattern, while the maternal morphometric parameters decreased as a result of the tumour’s progression. (Appendix A).

To understand the differences between the tumour-bearing dams (W) and the control group, and better comprehend the reductions in placental and foetal weights during the pregnancy evolution, we analysed the metabolic profile variations of serum, placental, and foetal tissues in these tumour-bearing dams, comparing them at three stages of gestation (12 dpc, 16 dpc, and 19 dpc).

### 2.3. Walker-256 Tumour Evolution Induced Changes Serum Metabolomic Profile

The tumour growth during pregnancy altered 42 out of 51 maternal serum metabolites identified (Appendix A). We found that 18 altered serum metabolites in the W group compared to C at 12 dpc; only tryptophan and uracil were increased (Figure 2p,r), while the other 16 metabolites were decreased (Figure 2, Appendix A). At 16 dpc, the tumour significantly increased the levels of 37 metabolites compared to the respective control group (Figure 2, Appendix A). At 19 dpc, we observed an increase in 18 metabolites in the presence of the tumour, whereas only aspartate levels were reduced (Figure 2c, Appendix A). Figure 2 displays the most significant changes in the metabolites. Methylhistidine, alanine, lactate, pyruvate and urea increased alongside tumour progression (Figure 2a,b,k, Appendix A), indicating the cancer growth’s harmful effects on the body’s wasting state. Therefore, to identify the tumour damage in the impacted metabolic vias, we analysed the changed metabolites in serum throughout the pregnancy. The results are presented in Table 1.

### 2.4. Walker-256 Tumour Evolution Induced Changes in Placental Metabolomic Profile

Among the total number of placental metabolites, we found at least 29 metabolites altered in the W group throughout the three gestational stages (Table 1, Figure 3, Appendix A). At 12 dpc, we found 23 altered placental metabolites, where, among these, lactate, succinate and 1-methylhistidine were increased (Figure 3a,g,l), while the other 20 metabolites were decreased in group W as compared to group C. At 16 dpc, 18 metabolites in the placenta changed in group W; among them, lactate and succinate were increased (Figure 3g,l), while 16 placental metabolites were reduced due to tumour growth during pregnancy. At 19 dpc, tumour growth during pregnancy resulted in alterations of 19 placental metabolites; 7 of these metabolites were decreased in group W, while 12 metabolites increased in group W. Regarding the tumour presence during pregnancy evolution, the increased levels of lactate and decreased glucose, as well as serine, were clearly related to energy dysfunction in the placenta of W (Figure 3f,g,k, Appendix A).

In order to identify how tumour development affects placental metabolic pathways, we analysed the variations in metabolite levels throughout pregnancy. The results are presented in Table 1.

### 2.5. Walker-256 Tumour Evolution Induced Changes in Foetal Metabolomic Profile

In particular, at 12 dpc, we identified 17 altered metabolites due to tumour growth during pregnancy; out of those, 7 metabolites increased (ATP, alanine, glutamine, lactate, lysine, NAD+, UDP-glucose; Figure 4b,e,g,j,k; Appendix A), and 10 metabolites were decreased in group W compared to group C (Appendix A). At 16 dpc, we found 10 altered metabolites in group W, with an increase in ADP, inosine, UDP-glucose and uridine content (Figure 4d,k,n), and 9 other metabolites decreased in response to tumour growth during pregnancy (Appendix A). At 19 dpc, we observed 15 altered foetal metabolites; out of these, 6 metabolites, including ADP, AMP, ATP, alanine, lactate and succinate showed an increase (Figure 4b–e,j,l), while 9 metabolites declined due to tumour evolution (Appendix A). Table 1 displays the primary metabolic pathways that were affected in the foetuses of the tumour-bearing dams.

## 3. Discussion

When a mother experiences pathophysiological changes like cancer, the placenta, may not be able to adapt metabolically, failing to maintain itself and provide optimal conditions for the healthy growth and development of the foetus. To the best of our knowledge, no previous publication has combined the analysis of the metabolomic profile of both the placenta and foetus in association with the maternal serum profile under tumour growth evolution. Hence, it is essential to conduct more studies to demonstrate the connection and significant metabolic pathways affected by cancer between the placental and foetal tissues. This work showed that neoplastic evolution causes changes in synthesis, degradation and energy sources vias during pregnancy, which opens up possibilities for identifying potential treatments that can ensure placental and foetal viability. Thus, the nuclear magnetic resonance technique was used to evaluate the altered serum, placental and foetal metabolites due to tumour growth throughout rat pregnancy in a time-course analysis (three stages of pregnancy), corresponding to similar stages of human pregnancy.

In this study, our focus was on the placental tissue, which is at the centre of the impact of tumour evolution, and we correlated the serum metabolic changes and their impact on foetal development. We analysed the time course of the rat pregnancy, knowing that at the middle of gestation (14th to 16th dpc), the placenta becomes fully functional, reaching its growth peak at 19 dpc, where placental weight remains stable [14,15,16]. Ensuring placental functions are crucial to guarantee optimal foetal growth and birth weight, which in turn plays a significant role in determining the foetus’s health [17]. Small-for-gestational-age foetuses have a higher risk of perinatal mortality and are more prone to future health complications such as obesity, diabetes, and cardiovascular, endocrine, and neural development disorders [18].

Maternal homeostasis changes can impact foetal weight, but placental insufficiency is considered a more significant factor, especially by the exchange maternal–foetal occurring mainly through the highly vascularized labyrinth zone [15,18]. Therefore, placental efficiency is related to the ability to transfer nutrients to the foetus, where the foetal weight/placenta ratio is considered an indicator of its efficiency [15]. As seen in this study, the presence of the tumour led to a reduction in foetal weight at 16 dpc and 19 dpc, as well as in foetal/placental ratio clearly corroborating with lower placental efficiency, as previously observed, a severe placental morphological change with increased oxidative stress and DNA fragmentation [8,19,20,21], suggesting a placental dysfunction [22,23,24].

Thus, changes in the metabolic profile of the placenta from tumour-bearing dams may indicate alterations in foetal growth. We noticed that changes in the metabolic profile of the serum could affect the metabolic pathways in the placenta, specifically those related to amino acids, carbohydrates, nucleotides, lipids, cofactors, vitamins and genetic translation. As a result, this could negatively impact placental function, leading to reduced foetal growth. At the beginning of the pregnancy (12 dpc) associated with the tumour, most of the metabolites related to amino acid metabolism were decreased in the serum and in the placental tissue of the tumour-bearing dams; this profile may be related to intense depletion due to tumour growth, offering fewer nutrients and substrate to the placenta’s establishment. Although, at the middle to the end of the pregnancy course, most of the serum amino acids content increased, showing a clear relation to the tumour-induced host spoliation. Throughout this process, the tumour was able to obtain nutrients from maternal tissues for its growth, but the placenta was unable to maintain its function probably due to a significant reduction in amino acids in the placental tissue. It is worth noting that some increased amino acids, like succinate, alanine, creatine and 1-methylhistidine, may be related to dysfunctional placental cell activity related to energy metabolism and protein degradation. This fact corroborates the compromised function of the placenta, as a study conducted in MAC16 tumour-bearing mice showed that the presence of the tumour during pregnancy negatively regulated the MAPK/ERK and PI3K/Akt/mTOR signalling pathways, resulting in a decrease in foetal growth and a decrease in placental cell activity [6]. Thus, it has been shown that tumour growth during pregnancy can alter the regulation of pathways responsible for amino acid transport through the placenta, which occurs through the activation of p38 MAPK signalling [25,26]. In addition, mTORC1 regulates placental amino acid transport by modulating the isoforms of the system A and system L transporters on the surface of trophoblasts [27,28]. Based on the changes in metabolites observed, the tumour growth during pregnancy was able to decrease the availability of nutrients or alter placental cell activity, leading to foetal impairment.

It was evident that the foetuses from tumour-bearing dams underwent spoliation, as there was a noticeable rise in lactate and succinate energy metabolites. Thus, the differences in these metabolites led to changes in glycolytic activity, the citric acid cycle, pyruvate metabolism and glyoxylate and dicarboxylate metabolic pathways. Glucose is the main energy substrate for both the placenta and foetus [29]. During early gestation, under constant growth and intense activity, the placenta uses approximately 50% of maternal circulating glucose; only 20% is directed to the foetus, and a small portion is converted to lactate [29]. At the end of gestation, when the foetus completes its development to ensure the organs and systems maturation, there is great mobilisation of maternal glucose to support placental activity, making it available to the foetus, which is in intense growth [29]. As the foetus has a limited ability to produce glucose for its own demands, the availability of glucose supply depends on the maternal circulation and the ability of the placenta to supply this glucose to the foetus [30]. Changes in placental energy metabolism can affect energy production for the placenta itself, as well as compromise energy production directed to the foetus, which implies adverse results like intra-uterine growth restriction and foetal hypoxia [31]. As seen here, in the tumour evolution in pregnant rats, the serum metabolite content was mainly related to the tumour-induced waste process, diverting subtracts to the gluconeogenesis process. The conversion of pyruvate to lactate by lactate dehydrogenase is normally present in placental energy metabolism, where about 30% of maternal circulating glucose is converted to lactate [29]. In the absence of lactate, the process of glycolysis is directed to the citric acid cycle, where NADH and FADH molecules are formed, in addition to increased ATP production. When most of the glucose is converted to lactate, instead of forming acetyl-CoA for the citric acid cycle, anaerobic glycolysis is the main process, leading to inefficient energy production [29]. Despite having no differences in placental intermediate molecules (such as ADP, AMP, ATP, NAD+ and FAD+), the high concentration of lactate found in the placentas of tumour-bearing rats, at 12 dpc, 16 dpc, and 19 dpc, suggests inefficiency in the conversion of pyruvate to acetyl-CoA, probably due to a reduction in the citric acid cycle, throughout pregnancy (Figure 3 and Appendix A), even though these placentas were able to modulate their energy metabolism in some way, maintaining concentrations of ADP, AMP, ATP, NAD+ and FAD+ at 16 dpc and 19 dpc. Thus, the placenta of tumour-bearing rats may fail in converting threonine and serine amino acids to pyruvate. This change in the normal citric acid cycle metabolism resulted in the placenta’s adaptation to alternative sources of energy.

Additionally, there was a decrease in the glucose amount in both the placental tissue and the developing foetus. This may suggest that the placenta was not providing adequate glucose supply to the foetus. The placenta has a high metabolic demand and needs a constant supply of energy. Creatine may be essential for ensuring there is enough ATP to provide the energy needs of the placental cells [32]. Here, we found in pregnant tumour-bearing dams lower levels of creatine in their placental tissue at 12 dpc; however, the levels increased at 19 dpc, which may be related to the higher ATP required to maintain the placental function. Corroborating, Ellery and colleagues found that during the 3rd trimester, foetuses with growth restriction had a 43% higher placental creatine content than control foetuses [33,34]. Likewise, as seen here, the increased placental creatine by 19 dpc could be related to placental insufficiency, compromising placental ATP homeostasis and thereby leading to lower development of foetuses in tumour-bearing dams [34]. In fact, the process of glycolysis in the placenta has a close connection with the pentose pathway and nucleotide metabolism [35]. This connection is crucial for the production of ribose-5-phosphate, which is used to create NADPH needed for RNA and DNA synthesis [36]. Furthermore, the regeneration of reduced glutathione from its oxidized form is dependent on NADPH, which is critical to cell antioxidant defences and crucial for the welfare of the foetus [36]. It is possible that the alteration of purine and pyrimidine pathways, which are involved in nucleotide metabolism, could occur due to impaired placental carbohydrate metabolism caused by tumour-induced wasting. This could lead to impaired foetal development, where the most significant impact was the alteration in the foetal purine pathway. Some purines play an important role in pathways that require energy, such as ATP, guanosine triphosphate (GTP), and the intracellular signalling pathways (cAMP and GMP) also playing key roles as cofactors, as nicotinamide adenine dinucleotide (NAD) and flavin adenine dinucleotide (FAD). Despite foetal glucose deficiency having been observed from 16 dpc to 19 dpc, it was found that at 12 dpc, there were higher amounts of altered metabolites associated with purine metabolism (Figure 4, Appendix A). It is possible that diseases occurring in the middle or late stages of pregnancy may cause less damage when the placenta is fully established. Furthermore, the changes in placental aminoacyl-tRNA biosynthesis due to the presence of the tumour could be related to the impairment of crucial translation processes, enzyme and protein biosynthesis, and also gene expression [37,38,39]. Here, we found that the development of a tumour caused a decrease in placental valine and pantothenate content by 12 dpc, which compromised the pantothenate pathway and CoA biosynthesis necessary for foetal development. Pantothenic acid deficiencies during embryogenesis in rats can lead to congenital malformation and retard foetal growth [40], which could explain a normal placental weight associated with a reduction in foetal weight by the 16 and 19 dpc in the tumour-bearing group. The effective supply of amino acids to the foetus is essential for intrauterine foetal growth [41,42]. On the other hand, the reduced concentration of placental choline and ethanolamine suggests a downregulation in glycerophospholipid metabolism at 19 dpc in the tumour-bearing dams. In intrauterine growth restriction, impaired placental transfer of lipophilic compounds (long-chain polyunsaturated fatty acids and lipophilic vitamins) appears to reinforce foetal metabolic dysfunction and decreased birth weight, as seen here [43]. Thus, our results suggest that the presence of the tumour during pregnancy not only causes inefficiency of placental glucose metabolism but also leads to a negative impact on amino acids and glycerophospholipid metabolism pathways, contributing to the reduction in foetal weight. It is still unclear whether the changes observed at 16 dpc and 19 dpc are the result of tumour growth during early pregnancy that may have affected the placenta, or if they are a direct result of the tumour growth itself. Additional research is required to ascertain the underlying reason for these changes. The conclusive confirmation of reduced placental function may be limited by the absence of placental histological assessment in the present work. However, we have previously published histological changes and some placental markers observed in the placentas of tumour-bearing rats, which led to impaired function. While this study offers valuable insights, it is important to acknowledge its limitations, particularly regarding preclinical trials. This work highlights specific alterations and unique responses related to the Walker-256 tumour, a rapidly growing tumour known to induce cachexia, limiting the translational clinical view. Our laboratory is currently conducting analyses to improve our understanding of the harmful effects caused by tumour evolution during pregnancy and find strategies to prevent or mitigate these effects, aiming to improve the well-being of the placenta and foetus during pregnancy affected by cancer.

## 4. Materials and Methods

### 4.1. Animals and Experimental Design

Forty-eight Wistar female rats, 90 days old, were obtained from the Multidisciplinary Centre for Biological Research in the Science of Laboratory Animals at UNICAMP (CEMIB-UNICAMP). The animals were kept in our own animal house under controlled temperature, humidity conditions and a light/dark cycle (12 h:12 h). According to the harem method, four females were housed with one male, and the 0-day post-conception (dpc) was considered when vaginal plugs or sperm were detected by vaginal smear. After that, the number of days post-conception was counted until pregnant female euthanasia.

The pregnant females were randomly assigned to two experimental groups: control (C) and Walker-256 tumour-bearing (W). (Figure 5). The animals were monitored daily, weighed three times/week, and fed a normoprotein diet, in accordance with the American Institute of Nutrition (AIN-93) [44], and water ad libitum.

The pregnant females were euthanised at three different stages of pregnancy: approximately on the 12th day (12 dpc), corresponding to the earlier time of the pregnancy (equivalent to the 1st trimester of human pregnancy); approximately on the 16th day (16 dpc), corresponding to the middle time of pregnancy (equivalent to the 2nd trimester of human pregnancy); and approximately on the 19th day (19 dpc), corresponding to the later stage of pregnancy (3rd trimester of human pregnancy). After euthanasia, serum was collected by heart puncture and prepared for metabolomic assays. Placental tissues (placenta with the maternal decidua) and foetuses were collected, weighed and prepared for metabolomic assays. Figure 5 shows the scheme of the experimental protocol.

### 4.2. Tumour Implant

A Walker-256 tumour cell suspension was implanted in the W group, by a single inoculation of 5 million viable neoplastic cells (using a saline solution), in approximately 0.3 mL of inoculum, injected in the subcutaneous tissue of the right flank [45]. The general guidelines of the UKCCCR (United Kingdom Coordinating Committee on Cancer Research 1988) for animal welfare were followed [46].

### 4.3. Sample Prepared for Metabolomic Assay

For the serum samples, 500 µL volume was filtered for 50 min at 13,000 rpm through a Microcon YM-3 column with a 3 kDa membrane (Amicon Ultra 0.5 mL, Sigma-Aldrich, Carrigtown, CO, USA). For the placental and foetal (portion from waist to hind legs) tissues, approximately 0.3 g from each was homogenised in 250 µL of a solution containing methanol and chloroform, respectively, at a ratio of 2:1, following the protocol proposed by Le Belle [47]. The detailed protocol for serum and tissue metabolite extraction and ^1^H-NMR Spectra Acquisition and Metabolic Quantification can be found in our previous studies [48].

### 4.4. Statistical Analysis

Morphometric parameters: The morphometric data are expressed as the means ± standard deviations (SD). The time-course comparison analyses were obtained using a one-way analysis of variance (ANOVA) with FDR-corrected *p*-values, followed by post hoc analyses using Tukey’s honestly significant difference (Tukey’s HSD), comparing the W vs. C groups and, for the same gestational period, were performed using Student’s t-test for probability < 5% [49], using Prism 7.0 software (Version 7.00, 31 March 2016).

^1^H NMR data analyses: All these analyses were performed using the online platform MetaboAnalyst 5.0 (a statistical, functional and integrative analysis of metabolomic data), showing the discrepancies between the identified metabolites and their concentrations among the experimental groups [50]. The principal component analysis (PCA) was implemented, as well as the partial least squares discriminant analysis (PLS-DA) models, and the data were extracted at a confidence level of 95% (Appendix A).

## Figures and Tables

**Figure 1 ijms-24-13026-f001:**
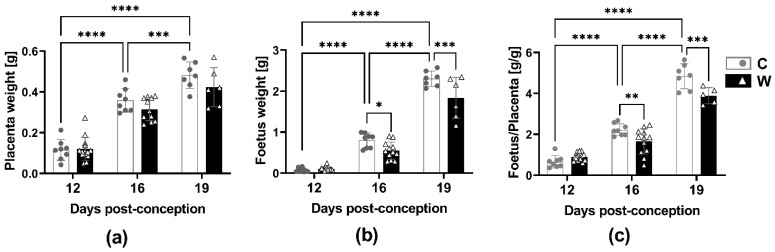
Placenta and foetus morphometric data from experimental groups, healthy (C) and tumour-bearing (W) dams at three distinct gestational periods, 12, 16, and 19 days post-conception: (**a**) placenta weight (g); (**b**) foetus weight (g); (**c**) ratio of foetal weight and placental weight. Data presented as mean ± standard deviation. Number of animals per group: C (placenta and foetus: 12 dpc *n* = 8; 16 dpc *n* = 8; 19 dpc *n* = 7); W group (placenta and foetus: 12 dpc *n* = 14; 16 dpc *n* = 12; 19 dpc *n* = 6). *, *p* < 0.05; **, *p* < 0.01; ***, *p* < 0.001; ****, *p* < 0.0001.

**Figure 2 ijms-24-13026-f002:**
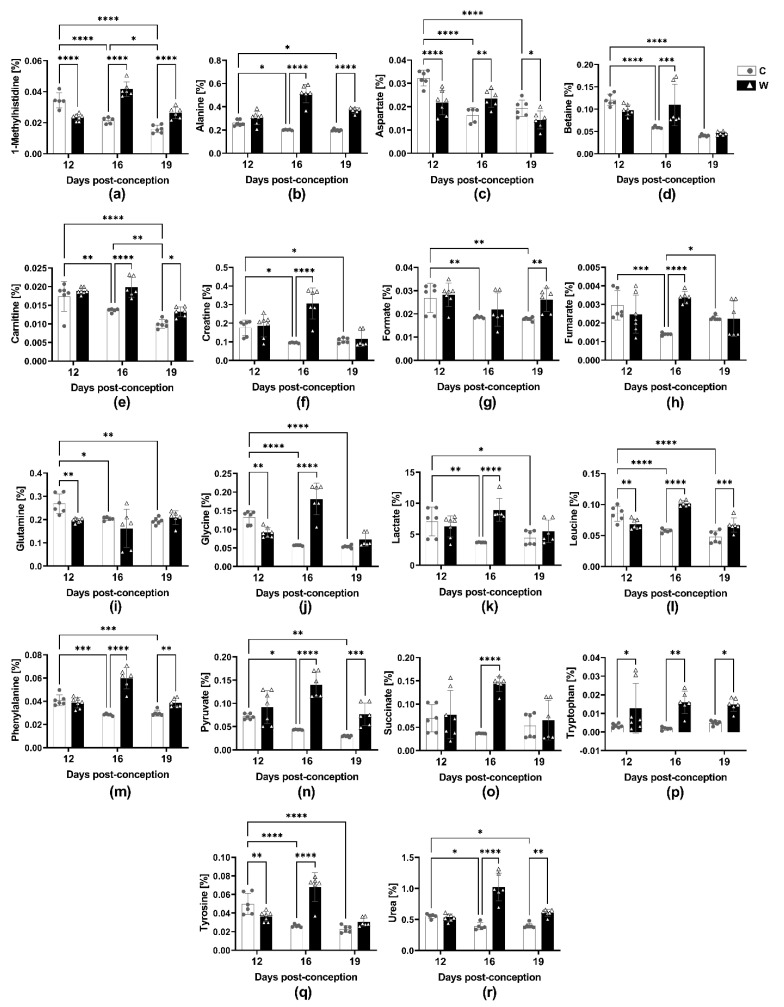
Maternal serum metabolite variations in healthy (C) and tumour-bearing (W) dams at three different gestational periods, 12-, 16- and 19-days post-conception. Number of animals per group: C at 12 dpc (*n* = 6), 16 dpc (*n* = 5) and 19 dpc (*n* = 6); W group at 12 dpc (*n* = 7), 16 dpc (*n* = 6) and 19 dpc (*n* = 6). For details see the Section 4. Legend: (**a**) 1-Methylhistidine; (**b**) Alanine; (**c**) Aspartate; (**d**) Betaine; (**e**) Carnitine; (**f**) Creatine; (**g**) Formate; (**h**) Fumarate; (**i**) Glutamine; (**j**) Glycine; (**k**) Lactate; (**l**) Leucine; (**m**) Phenylalanine; (**n**) Pyruvate; (**o**) Succinate; (**p**) Tryptophan; (**q**) Tyrosine; (**r**) Urea. The data were presented as the mean ± standard deviation. Statistical analysis was performed using a two-way ANOVA, and multiple comparisons were corrected using the post hoc test, specifically the multiple comparisons test. *, *p* < 0.05; **, *p* < 0.01; ***, *p* < 0.001; ****, *p* < 0.0001.

**Figure 3 ijms-24-13026-f003:**
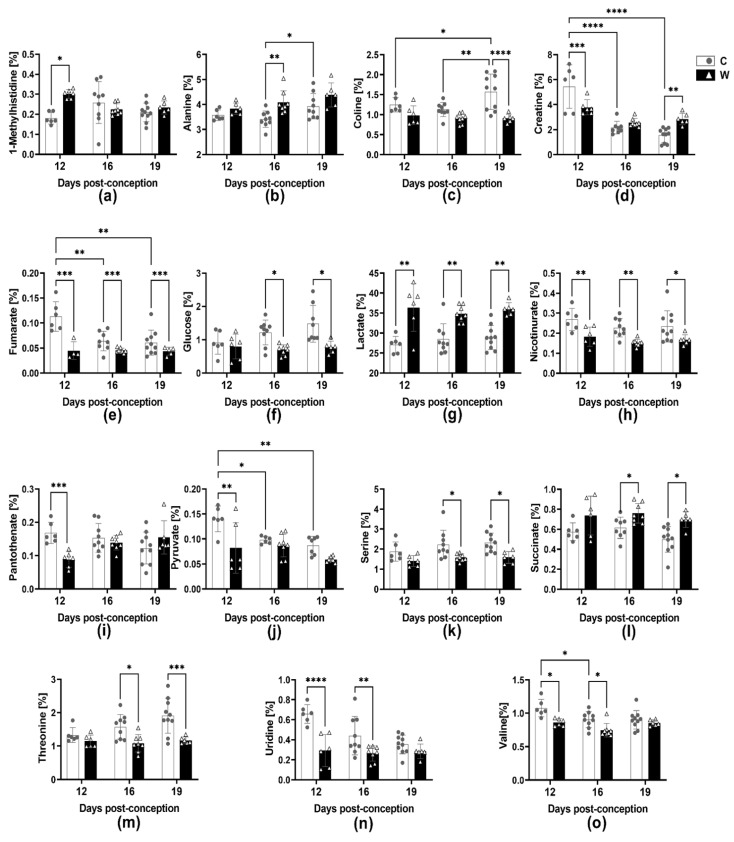
Placental metabolite variations in healthy (C) and tumour-bearing (W) dams at three different gestational periods, 12, 16 and 19 days post-conception. Number of animals per group: C at 12 dpc (*n* = 6), 16 dpc (*n* = 9) and 19 dpc (*n* = 6); W group at 12 dpc (*n* = 6),16 dpc (*n* = 8) and 19 dpc (*n* = 6). Legend: (**a**) 1-Methylhistidine; (**b**) Alanine; (**c**) Choline; (**d**) Creatine; (**e**) Fumarate; (**f**) Glucose; (**g**) Lactate; (**h**) Nicotinurate; (**i**) Pantothenate; (**j**) Pyruvate; (**k**) Serine; (**l**) Succinate; (**m**) Threonine; (**n**) Uridine; (**o**) Valine. The data were presented as the mean ± standard deviation. Statistical analysis was performed using a two-way ANOVA, and multiple comparisons were corrected using the post hoc test, specifically, the multiple comparisons test. Pregnancy: *, *p* < 0.05; **, *p* < 0.01; ***, *p* < 0.001; ****, *p* < 0.0001.

**Figure 4 ijms-24-13026-f004:**
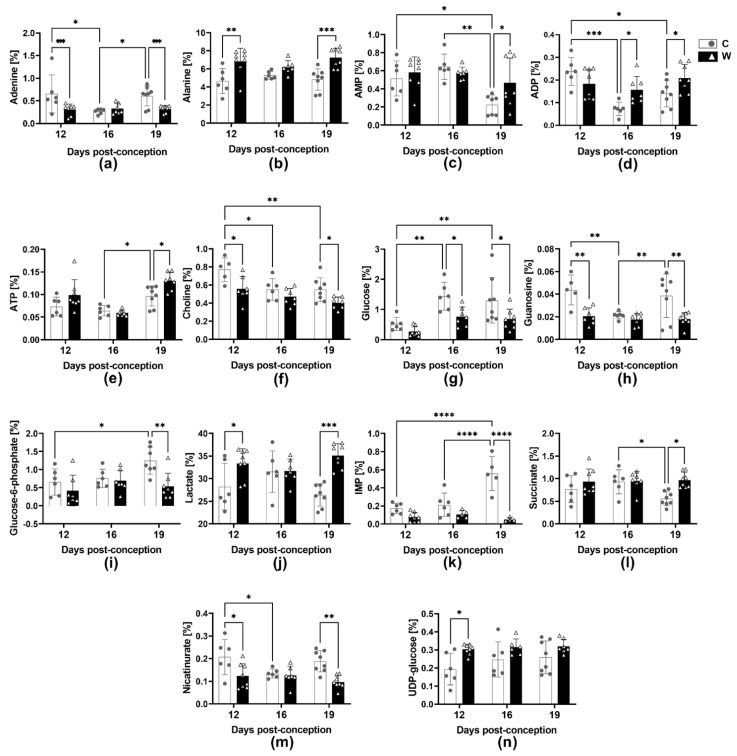
Foetal metabolite variations in healthy (C) and tumour-bearing (W) dams at three different gestational periods, 12, 16 and 19 days post-conception. Number of animals per group: C at 12 dpc (*n* = 6), 16 dpc (*n* = 6) and 19 dpc (*n* = 8); W group at 12 dpc (*n* = 8), 16 dpc (*n* = 7) and 19 dpc (*n* = 8). Legend: (**a**) Adenine; (**b**) Alanine; (**c**) AMP; (**d**) ADP; (**e**) ATP; (**f**) Choline; (**g**) Glucose; (**h**) Guanosine; (**i**) Glucose-6-phosphate; (**j**) Lactate; (**k**) IMP; (**l**) Succinate; (**m**) Nicotinurate; (**n**) UDP-Glucose. The data were presented as the mean ± standard deviation. Statistical analysis was performed using a two-way ANOVA, and multiple comparisons were corrected using the post hoc test, specifically the multiple comparisons test. *, *p* < 0.05; **, *p* < 0.01; ***, *p* < 0.001; ****, *p* < 0.0001.

**Figure 5 ijms-24-13026-f005:**
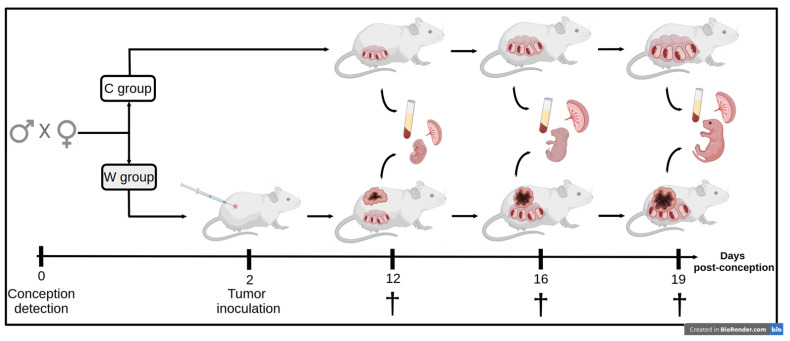
Experimental procedure to analyse serum, placental and foetal metabolomic profiles in pregnant tumour-bearing rats (W) compared to the healthy pregnant group (C). All the animals were euthanised at 12, 16 and 19 days of pregnancy. Illustration created using BioRender (accessed on 19 July 2023).

**Table 1 ijms-24-13026-t001:** The impacted metabolic pathways in serum, placenta and foetus in tumour-bearing dams compared with the control group at three different pregnancy stages.

		W vs. C								
		Serum			Placenta			Foetus		
		12 dpc	16 dpc	19 dpc	12 dpc	16 dpc	19 dpc	12 dpc	16 dpc	19 dpc
Genetic Processing—Translation	**Aminoacyl-tRNA Biosynthesis**									
	Alanine	-	⇑	⇑	-	⇑	⇑	-	-	-
	Arginine	⇓	-	-	-	-	-	-	-	-
	Asparagine	⇓	⇑	⇑	-	-	-	-	-	-
	Aspartate	⇓	⇑	⇓	-	-	-	-	-	-
	Glutamate	-	-	-	-	-	-	-	-	-
	Glutamine	⇓	-	-	⇓	-	⇑	-	-	-
	Glycine	⇓	⇑	-	-	-	-	-	-	-
	Histidine	⇓	⇑	-	-	-	-	-	-	-
	Isoleucine	⇓	⇑	⇑	⇓	-	-	-	-	-
	Leucine	⇓	⇑	⇑	-	-	-	-	-	-
	Lysine	-	⇑	⇑	-	-	-	-	-	-
	Methionine	-	⇑	-	⇓	-	⇓	-	-	-
	Phenylalanine	-	⇑	⇑	-	⇓	-	-	-	-
	Proline	⇓	⇑	⇑	-		-	-	-	-
	Serine	-	-	-	-	⇓	⇓	-	-	-
	Threonine	-	-	-	-	⇓	⇓	-	-	-
	Tryptophan	⇑	⇑	⇑	⇓		-	-	-	-
	Tyrosine	⇓	⇑	-	-	⇓	-	-	-	-
	Valine	-	⇑	⇑	⇓	⇓	-	-	-	-
Amino acid metabolism	**Alanine, Aspartate and Glutamate Metabolism**									
	Alanine	-	⇑	⇑	-	⇑	⇑	⇑	-	⇑
	Asparagine	⇓	⇑	⇑	-	-	-	-	-	-
	Aspartate	⇓	⇑	⇓	-	-	-	-	-	-
	Fumarate	-	⇑	-	⇓	-	-	⇓	-	-
	Glutamate	-	-	-	-	-	-	-	-	-
	Glutamine	⇓	-	-	⇓	⇓	⇓	⇑	-	-
	Pyruvate	-	⇑	⇑	⇓	-	-	⇓	-	-
	Succinate	-	⇑	-	⇑	⇑	⇑	-	-	⇑
	**Arginine Biosynthesis**									
	Arginine	⇓	⇑	-	-	-	-	-	-	-
	Aspartate	⇓	-	⇓	-	-	-	-	-	-
	Fumarate	-	⇑	-	⇓	-	-	⇓	-	-
	Glutamate	-	-	-	-	-	-	-	-	-
	Glutamine	⇓	-	-	⇓	-	-	⇑	-	-
	Ornithine	-	⇑	⇑	-	-	-	-	-	-
	Urea	-	⇑	⇑	-	-	-	-	-	-
	**Histidine Metabolism**									
	1-Mehyl-histidine	⇓	⇑	-	⇑	-	-	-	-	-
	Aspartate	⇓	⇑	-	-	-	-	-	-	-
	Glutamine	-	-	-	-	-	-	-	-	-
	Histamine	-	⇑	-	⇓	-	-	-	-	-
	Histidine	⇓	⇑	-	-	-	-	-	-	-
	**Glycine, Serine and Threonine Metabolism**									
	Betaine	-	⇑	-	-	-	-	-	-	-
	Choline	-	⇑	-	-	-	⇓	⇓	-	⇓
	Creatine	-	⇑	-	⇓	-	⇑	-	-	⇓
	Glutamine	-	-	-	-	-	-		-	-
	Glycine	-	⇑	-	-	-	-	-	-	-
	Guanidoacetate	-	-	-	⇓	-	-	-	-	-
	N,N-DG	-	⇑	-	-	-	-	-	-	-
	Pyruvate	-	⇑	-	⇓	-	-	⇓	-	-
	Serine	-	-	-	-	⇓	⇓	-	-	-
	Threonine	-	-	-	-	⇓	⇓	-	-	-
	**Arginine and Proline Metabolism**									
	Arginine	-	-	-	-	-	-	-	-	-
	Creatine	-	-	-	⇓	-	-	-	-	-
	Glutamine	-	-	-	-	-	-	-	-	-
	Guanidoacetate	-	-	-	⇓	-	-	-	-	-
	Ornitine	-	-	-	-	-	-	-	-	-
	Proline	-	-	-	-	-	-	-	-	-
	Pyruvate	-	-	-	⇓	-	-	-	-	-
	**Valine, Leucine and Isoleucine Biosynthesis**									
	Isoleucine	⇓	⇑	⇓	⇓	-	-	-	-	-
	Leucine	⇓	⇑	⇓	-	-	-	-	-	-
	Proline	-	-	-	-	-	-	-	-	-
	Threonine	-	-	-	-	⇓	-		-	-
	Valine	⇓	⇑	⇓	⇓	⇓	-	-	-	-
	**Phenylalanine, Tyrosine and Tryptophan Biosynthesis**									
	Phenylalanine	-	⇑	-	-	⇓	-	-	-	-
	Tyrosine	-	⇑	-	-	⇓	-	-	-	-
Carbohydrate metabolism	**Glyoxylate and Dicarboxylate Metabolism**									
	Acetate	⇓	-	-	⇓	-	-	-	-	-
	Formate	-	-	-	-	-	-	-	-	-
	Glutamate	-	-	-	-	-	-	-	-	-
	Glutamine	⇓	-	-	⇓	⇓	⇓	⇑	-	-
	Glycine	⇓	-	-	-	-	-	-	-	-
	Pyruvate	-	-	-	⇓	−	−	⇓	-	-
	Serine	-	-	-	-	⇓	⇓	-	-	-
	Uracil	-	-	-	-	-	-	-	-	-
	Valine	-	-	-	-	-	-	-	-	-
	**Pyruvate Metabolism**									
	Acetate	-	-	-	⇓	-	-	-	-	-
	Fumarate	-	-	-	⇓	-	-	⇓	-	-
	Lactate	-	-	-	⇑	-	-	⇑	-	-
	Pyruvate	-	-	-	⇓	-	-	⇓	-	-
	**Glycolysis/Gluconeogenesis**									
	Acetate	-	-	-	⇓	-	-	-	-	-
	Glucose	-	-	-	-	⇓	⇓	-	⇓	⇓
	G-6-P	-	-	-	-	-	-	-	-	⇓
	Lactate	-	-	-	⇑	⇑	⇑	⇑	-	⇑
	Pyruvate	-	-	-	⇓	-	-	⇓	-	-
	**Citrate Cycle (TCA Cycle)**									
	Fumarate	-	-	-	⇓	-	-	⇓	-	-
	Pyruvate	-	-	-	⇓	-	-	⇓	-	-
	Succinate	-	-	-	⇑	-	-	-	-	-
Metabolism of cofactors and vitamins	**Pantothenate and CoA biosynthesis**									
	Aspartate	⇓	⇑	-	-	-	-	-	-	-
	Pantothenate	⇓	⇑	-	⇓	-	-	-	-	-
	Uracil	⇑	⇑	-	-	-	-	-	-	-
	Valine	-	⇑	-	⇓	-	-	-	-	-
Nucleotide metabolism	**Purine Metabolism**									
	Adenine	-	-	-	⇓	-	⇓	⇓	-	⇓
	Adenosine	-	-	-	⇓	⇓	-	⇓	-	-
	ADP	-	-	-	-	-	-	-	⇑	⇑
	AMP	-	-	-	-	-	-	-	-	⇑
	ATP	-	-	-	-	-	-	⇑	-	⇑
	Glutamine	-	-	-	⇓	⇓	⇓	⇑	-	-
	GTP	-	-	-	⇓	-	⇓	⇓	-	-
	Guanosine	-	-	-	-	-	-	⇓	-	⇓
	IMP	-	-	-	-	-	-	⇓	⇓	⇓
	Inosine	-	-	-	⇓	⇓	⇓	⇓	-	⇓

Legend: Pregnant healthy (C) and tumour-bearing (W) groups. Serum, placental, and foetal impacted metabolic pathways. *p* value (FDR) < 0.05 for ⇑, increased; ⇓, decreased. -, unchanged metabolite.

## Data Availability

https://osf.io/j9brz/?view_only=6a07ef6fc5ab44c398d629645ae6d852 (accessed on 19 July 2023).

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
