# Peer review of "Placental, Foetal, and Maternal Serum Metabolomic Profiles in Pregnancy-Associated Cancer: Walker-256 Tumour Model in a Time-Course Analysis"

_ijms, 2023, doi:10.3390/ijms241713026_

Round 1

Reviewer 1 Report

The Authors aimed to describe the impact of tumor evolution in serum, placental and fetal metabolomics profiles during pregnancy. By using a rat model, they reported a lower speed in placental growth and a decreases fetal weight. Also, serum metabolomic changes related to tumor growth were observed in placental metabolomic, which directly harmed the fetal development.

The study has a good research question, methods are reported clearly to be replicated, and results are impressive. The paper is well-written, with a good readability.

Reorganization of the results might improve the structure of the manuscript, postponing results to methods. Additionally, to move some results from Supplemental materials to paper could be enrich the text and help the reader in better understanding the research. As limitation, to add the missing info of placental histology, able to confirm the reduced placental function.

Author Response

 We would like to express our thanks to Reviewer #1 for the time taken to analyse our manuscript and for providing comments and suggestions. We have full confidence that your insightful input has led to significant enhancements in this revised version of the manuscript.

Reorganization of the results might improve the structure of the manuscript, postponing results to methods.

               Answer: Thank you for your feedback, but we followed the journal rules where the material and methods should be placed after the results section.

Additionally, to move some results from Supplemental materials to paper could be enrich the text and help the reader in better understanding the research. 

               Answer: Thank you for your suggestion. Although, we choose to present some results as supplement material because of the size limitation as proposed by the journal rules.

As limitation, to add the missing info of placental histology, able to confirm the reduced placental function.

               Answer: We would like to thank for this suggestion and we have included some information about the placenta histology as a limitation of the present manuscript; we also included that the changes in histology and increased oxidative stress and DNA fragmentation observed in the placenta of tumour-bearing rats led to an impaired function as published in our previously works (Toledo and Gomes-Marcondes Oncol Res 1999, 11 (8), 359–366; Toledo,  et al. Placenta 2011, 32 (11), 859–864. https://doi.org/10.1016/j.placenta.2011.08.009 )  ( lines 215 to 218, references 19 and 20).

Reviewer 2 Report

Dear Authors,

This study is well conducted in terms of result conclusions and discussion. Although ending in the abstract conclusion in line 23 "jeoparded the concept" sounds a little bit weird as a reviewer, better rephrase into  "jeoparded the early embryonic development. The limitations are clearly written and I'm curious which additional analysis will contribute in unraveling this cancer pathophysiology in placental tissue and embryonic defects? The technical approach by using H-NMR analysis is challenging, since it took me more time to understand the table 1 content and the supplementary files (going back to your publication in Metabolites in 2021), but maybe there is a way to modulate these results in a biological model using all kind of parameters, which could lead to better biological understandings in studying these harmfully defects in future.

Kind regards, Reviewer

Author Response

We would like to thank Reviewer#2 for taking time to analyse our work and providing us with constructive feedback and suggestions to improve the manuscript.

This study is well conducted in terms of result conclusions and discussion. Although ending in the abstract conclusion in line 23 "jeoparded the concept" sounds a little bit weird as a reviewer, better rephrase into "jeoparded the early embryonic development.

               Answer: We would like to thank you for this point and we rephrased the conclusion adding as your suggestion: “jeoparded the early embryonic development leading to failure to foetal functions”.

The limitations are clearly written and I'm curious which additional analysis will contribute in unraveling this cancer pathophysiology in placental tissue and embryonic defects? 

               Answer: We thank for this comment, so we have revised the final statement in the discussion section to reflect our commitment to conducting new experiments aimed at improving our understanding of how to minimize the effects of tumour on foetal and placental tissues, aiming to find a therapeutic approach.

The technical approach by using H-NMR analysis is challenging, since it took me more time to understand the table 1 content and the supplementary files (going back to your publication in Metabolites in 2021), but maybe there is a way to modulate these results in a biological model using all kind of parameters, which could lead to better biological understandings in studying these harmfully defects in future.

               Answer: We agreed with your comments to investigate the impact of tumour effect on placental tissue through biological events, so we are now undergoing new experimental assays to analyse the changes in specific genes or cell signalling. Our aim is to utilise these changed metabolomic profiles to gain a better understanding of the association between cancer and pregnancy and then find how to improve or minimize the tumour damage effect over placenta and foetus. We also agree with this point and have rearranged the table to only show the comparison of the data from tumour-bearing rats with the control group. The control group data has been included in the supplementary table. We hope this new version of the table is clearer.